# The Optimization of Bayesian Extreme Value: Empirical Evidence for the Agricultural Commodities in the US

**Jittima Singvejsakul** [1,*], **Chukiat Chaiboonsri** [2] and **Songsak Sriboonchitta** [2,3]

1. Department of Agricultural Economy and Development, Faculty of Agriculture, Chiang Mai University, Chiang Mai 50200, Thailand
2. Faculty of Economics, Chiang Mai University, Chiang Mai 50200, Thailand; chukiat1973@gmail.com (C.C.); songsakecon@gmail.com (S.S.)
3. Puey Ungphakorn Center of Excellence in Econometrics, Chiang Mai University, Chiang Mai 50200, Thailand
* Correspondence: jittima.s@cmu.ac.th

**Abstract:** Bayesian extreme value analysis was used to forecast the optimal point in agricultural commodity futures prices in the United States for cocoa, coffee, corn, soybeans and wheat. Data were collected daily between 2000 and 2020. The estimation of extreme value can be empirically interpreted as representing crises or unusual time series trends, while the extreme optimal point is useful for investors and agriculturists to make decisions and better understand agricultural commodities future prices warning levels. Results from the Non-stationary Extreme Value Analysis (NEVA) software package using Bayesian inference and the Newton-optimal methods provided optimal interval values. These indicated extreme maximum points of future prices to inform investors and agriculturists to sell the contract and product before the commodity prices dropped to the next local minimum values. Thus, agriculturists can use this information as an advanced warming of alarming points of agricultural commodity prices to predict the efficient quantity of their agricultural product to sell, with better ways to manage this risk.

**Keywords:** agricultural commodity future prices; extreme value; NON-stationary Extreme Value Analysis (NEVA); Newton-optimal method

## 1. Introduction

Commodity futures are one of the most important asset classes. Investors are now increasingly investing their portfolios into commodity futures after the equity market crash in 2000. Investment in commodities is attractive in terms of diversification with respect to fixed income from equities that follows changes in inflation rates. The large changes in commodity prices that occurred in late 2008 have attracted considerable research attention. The financial crisis in 2008, caused by the subprime crisis in the United States, played a very important role in the economic system. Agricultural commodities are an influential group in the futures market and also impact economic productive growth in every country. Food price volatility is a critical problem for governments and regulators worldwide as most nations trade in food. High food prices can lead to poverty and malnourishment, especially in developing countries. The United Stated performs best farming practices and is the world's richest agricultural nation. The largest crops grown in the United States are corn and soybeans, with wheat, coffee and cocoa as the second rank of production. The agricultural industry in the United States contributes more than 100 USD billion to the economy, but agricultural exports remained depressed in 2019. Extreme price changes have become increasingly interesting in financial markets for many agricultural commodities. Commodity markets are widely used for risk management by producers and by the federal crop insurance program that has revenue protection. Extreme price events can have major implications on producer profitability. Consequently, futures prices for cocoa, coffee, corn, soybeans and wheat, as a broad range of agricultural commodities, are employed to

statistically investigate the optimal point in future prices to try to understand what could trigger the next global crisis and when. Two types of maximum points are defined as the global maximum and local maximum. The global maximum is the absolute maximum for the overall number of sets across the entire domain of the function, while the local maximum is the relative maximum from a particular neighborhood and might have many points for one set.

This paper focused on the estimation of the local maximum which, using the Non-stationary Extreme Value Analysis (NEVA) software package with Bayesian inference and the Newton-optimal methods, provided optimal interval values. Estimation of the optimal local maximum point is useful for investors and agriculturists to plan their investments and initiate product sales before the agricultural commodity future prices drop to the next local minimum point. The study of early warning points would be useful in a global financial crisis and commodity prices, especially in the agriculture sector. This study consists of five sections. After the introduction, Section 2 provides an overview as a literature review that includes Bayesian and other extreme value applications. Section 3 details the research methodology used to quantify the Bayesian extreme values for agricultural commodities, with results of our empirical analysis presented in Section 4, while key conclusions are drawn in Section 5.

## 2. Literature Review

This section provides an overview of previous research on extreme value applications. Historically, several studies have considered and investigated extreme value theory in an economic system. Gilli and Këllezi (2006) studied the measurement of financial risk using extreme values, focusing on the computation of tail risk measures and confidence intervals in the stock market index. Estimations showed a 0.01 probability that the loss value would be 2.397%, with the confidence interval finite at $1/0.671 > 1$. Peruvian stock market returns were studied by Gabriel in 2017 (Gabriel 2017) by investigating daily data to obtain 'value at risk' and 'expected shortfall' using the generalized Pareto distribution (GPD). Results showed a negative return from the stock market at 12.44% in 2011 and instability for the negative stock market from estimation of the Hill tail index. Oordt, Stork and Vries (Oordt et al. 2013) studied agricultural commodities extreme price risk. They showed that the agricultural price had a fat tail and occurred endogenously as a result of productivity shock for commodities futures in the United States as corn, oats, soybeans, wheat, cotton, sugar, orange juice, live cattle and lean hogs. Estimation of the extreme value of commodities in agriculture was also undertaken by Fretheim and Kristiansen (2015). They applied the extreme value theory approach to commodity market risk from 1995 to 2013 using commodity prices of corn, wheat, soybeans, soy oil, cocoa, orange juice, lean hogs and feeder cattle. Their contribution as an empirical analysis confirmed a well-established fact, namely, that the distribution of commodity price returns is fat-tailed relative to the Gaussian distribution. An analysis of the estimated shape parameters of the generalized extreme value (GEV) distribution further substantiated no systematic change in the extreme risk associated with commodity investments. They concluded that commodity return distributions had heavy tails during the period 1995 to 2013.

One statistical analysis is called the Bayesian extreme value approach. This method is proposed as an alternative analysis using Bayesian extreme values that can provide accurate results using random parameters, while the extreme value approach presents only fixed parameters. Several studies have investigated this method. For instance, Merwe, Steven and Pretorius (Merwe et al. 2018) explored the Bayesian extreme value analysis of stock exchange data, focusing on the fitting of the GPD beyond a threshold and improved the Bayesian methods with parameter estimation, while Wannapan, Chaiboonsri and Sriboonchitta (Wannapan et al. 2018) considered extreme values for macro-econometric forecasting of the gross domestic product (GDP), consumer price index (CPI) and foreign direct investment (FDI) using Non-stationary Extreme Value Analysis (NEVA) by applying Bayesian inference. Results showed extreme points of macroeconomic data that presented

as the interval value and optimal point. Park and Maples (2018) studied extreme events and the serial dependence of agricultural commodity prices. They used the daily price of five agricultural commodities as corn, soybeans, wheat, cotton and live cattle. Results revealed that the prediction accuracy of the Bayesian hierarchical model for serially dependent extremes outperformed the other candidates in measuring extreme risks in the agricultural commodity markets. The model captured the changes in the shape of a heavy-tailed distribution when calculating risk measures such as the expected price shortfall and value at risk (VaR).

However, limited studies exist concerning Non-stationary Extreme Value Analysis (NEVA) that apply Bayesian inference with financial data and commodity prices, especially agricultural commodities. Therefore, in this paper, we proposed the Bayesian extreme value using the Newton optimization processing method to study the extreme value point in five major agricultural commodities future prices in the United States including cocoa, coffee, corn, soybeans and wheat.

## 3. Research Methodology

In this study, several methods were employed to determine the forecast of Bayesian extreme value optimization in agricultural commodity future prices. Firstly, the augmented Dickey–Fuller (ADF) unit root test, based on Bayesian inference, was used to classify the stationary data and non-stationary data. Second, the Non-stationary Extreme Value Analysis (NEVA) method was employed to determine the extreme interval for both the non-stationary and stationary data. Lastly, the results from the NEVA were plugged into the random variable method to obtain a finite random set, before estimation using the Newton-optimal processing method to determine the optimal extreme point.

### 3.1. The Unit Root Test Using Bayesian

The unit root test is investigated by using the ADF test, which shows the ratio between the stationary data and non-stationary data of the null hypothesis (Said and Dickey 1984). The significant statistical issues associated with the autoregressive unit root test (AR) are defined as

$$x_t = c + \rho x_{t-1} + \varepsilon_t, \varepsilon_t \sim N(0, \sigma^2), \tag{1}$$

The prior density of $\rho$ is formulated and expressed as following:

$$p(\theta) = p(\phi)p(a^*|\phi), \tag{2}$$

The marginal likelihood for $\phi$ is

$$l(\phi|D)\alpha \int l(\rho|D)\phi(a^*|\phi)da^*, \tag{3}$$

The consideration of the hypotheses of Bayesian is combined with the Bayes factor to interpret the hypothesis of stationary data. The null hypothesis is defined by $N_i$ and the alternative hypothesis is denoted by $N_j$. The ratio of posterior odds of $N_i$ and $N_j$ is

$$\frac{p(N_i|y)}{p(N_j|y)} = \frac{p(y|N_i)}{p(y|N_j)} \times \frac{\pi(N_i)}{N_j}, \tag{4}$$

The interpretation in the Bayes factor can be interpreted in Table 1.

**Table 1.** The explanation of Bayes factor of Jeffrey Guideline model.

| Items | The Interpretation |
|-------|--------------------|
| BF < 1/10 | Strong evidence for $N_j$ |
| 1/10 < BF < 1/3 | Moderate evidence for $N_j$ |
| 1/3 < BF < 1 | Weak evidence for $N_j$ |
| 1 < BF < 3 | Weak evidence for $N_i$ |
| 3 < BF < 10 | Moderate evidence for $N_i$ |
| 10 < BF | Strong evidence for $N_i$ |

*3.2. The Generalized Pareto Distributions (GPD)*

The extreme event distributions for the threshold under condition (Pickands 1975) sing GPD to estimate the distribution as

$$G(x|\zeta, \sigma, u) = \begin{cases} 1 - \left(1 + \frac{\zeta(x-u)}{\sigma}\right) - 1/\zeta, & \text{if } \zeta \neq 0 \\ 1 - \exp\left[-\frac{(x-u)}{\sigma}\right], & \text{if } \zeta = 0 \end{cases}, \tag{5}$$

where $\sigma > 0$ and $\zeta$ are the scale and shape parameter, respectively.

Then, the threshold from the GPD equation is assumed to be the observations under the threshold u. Thus, the model $H(.|\eta)$ from the generation of u from a certain distribution with parameters $\eta$ is shown as

$$F(x|\eta, \zeta, \sigma, u) = \begin{cases} H(x|\eta), & \text{if } x < u \\ H(x|\eta) + [1 - H(x|\eta)]G(x|\zeta, \sigma, u), & \text{if } x \geq u \end{cases} \tag{6}$$

The likelihood function can be expressed as

$$L(\theta; x) = \prod_A h(x|\eta) \prod_B (1 - H(u|\eta)) \left\{ \frac{1}{\sigma}\left[1 + \frac{\zeta(x_i - u)}{\sigma}\right]_+^{-(1+\zeta)/\zeta} \right\}, \text{ for } \zeta \neq 0,$$

and

$$L(\theta; x) = \prod_A h(x|\eta) \prod_B (1 - H(u|\eta))[(1/\sigma)\exp\{(x_i - u)/\sigma\}], \text{ for } \zeta = 0, \tag{7}$$

The threshold (u) is the discontinuous density that depends on jumped density. The jumped distribution consists of small and large jumped density, of which the large jumps are more favorable than small jumps in terms of parameter estimation.

*3.3. The Non-Stationary Extreme Value Theory*

The Non-Stationary Extreme Value Analysis (NEVA), is proposed by Cheng et al. in 2014 (Cheng et al. 2014). This method uses Bayesian inference for the GPD of the distribution for both stationary and non-stationary conditions from which the posterior distribution of parameters is obtained by the Bayesian technical. The Markov chain Monte Carlo (MCMC) approach is used in several investigations of extremes for the arbitrary distribution. Thus, the GPD parameters of Bayes' theorem under the non-stationary assumption can be shown as

$$p\left(\beta|\vec{y}, x\right) = \prod_{N_t, t=1} p\left(\vec{y}|\beta, x(t)\right) = \prod_{N_t, t=1} p\left(\vec{y}|\mu(t), \sigma, \zeta,\right) \tag{8}$$

where $\beta = (\mu_1, \mu_0, \sigma, \zeta)$.

The stationary term can be shown as

$$p\left(\theta|\vec{y}\right)\alpha p\left(\vec{y}|\theta\right)p|(\theta) = \prod_{N_t, t=1} p(y_t|\theta)p(\theta) \tag{9}$$

where $\theta = (\mu, \sigma, \zeta)$.

The estimation of NEVA from the joint posterior distribution creates the number of realization using the differential evolution Monte Carlo Metropolis transitions (DE-MCMC). The individual $C_i$ can be described as $s_i$ for the non-dominated and $s_j$ for the dominated as

$$\text{fit}(C_i) = \begin{cases} s_i \\ 1 + \sum_i^{i=j} s_j \end{cases},$$

(10)

The MCMC is employed to create new samples with probabilities based on the variation of the $\text{fit}(C_i)$ value as

$$w\left(C_i \to C_t^i\right) = e^{\frac{-(\text{fit}(c_t')\text{fit}(c_t))}{T}}$$

(11)

where T is the simulation of the future prices, of which the generated values from $C_t'$ are accepted with the probability as

$$\min\left(1, w\left(c_i \to c_t'\right)\right)$$

(12)

*3.4. Random Sets*

The estimation from the NEVA is provided as an interval number. Then, the interval number is used to generate the random number before applying the Newton method to find the optimal point. The sampling random number is generated from the power set of $2^U$, which is interested in which U is a finite number (Polyak 2007). The set of $2^U$ is chosen by a probability density function $f : 2^U$ is $[0, 1], \sum_{A \subseteq U} f(A) = 1$. The coverage function is used to define the degree of separation as

$$\pi_s : U \to [0, 1], \ \pi_s(u) = P(u \in S) = \sum_{u \in A} f(A)$$

(13)

*3.5. The Newton Optimization Approach*

This approach finds the extreme value point, which is the process after investigating the sampling number. Thus, Newton's method, theorized by Newton in 1669, is used to estimate the optimal point of agricultural commodity future prices in the US. The function of the Newton method can be written as

$$F(x) = x_0$$

(14)

where $x_0$ is the initial point or made to be the starting point, and to assemble the linearapproximation of $F(x)$ in the neighbor of $x_0 : F(x_0) + F(x_0)h = 0$.

Then the calculation of the linear equation can be shown as

$$x_{k+1} = x_k F'(x_k)^{-1} F(x_k), k = 0, 1, \dots\dots$$

(15)

Newton's method proposes two approaches—the continuously differentiable and twice differentiable on the data—that can be expressed as follows:

$$|x_k - x^*| \leq \frac{\eta}{h2^k}(2h)^{2^k}$$

(16)

and

$$|x_k - x^*| \leq \frac{\beta\eta \pm h/2^{2^k-1}}{1 - h/2^{2^k}}$$

(17)

The calculation of Newton's method $F'(x)$ provides the tangent line at $x^{(0)}$ and root $x^{(1)}$, to show the calculation of the true $x^*$. Then, the next calculation of $x^{(2)}$ is produced following the same step as before and shows the root at $x^*$ that can be seen in Figure 1.

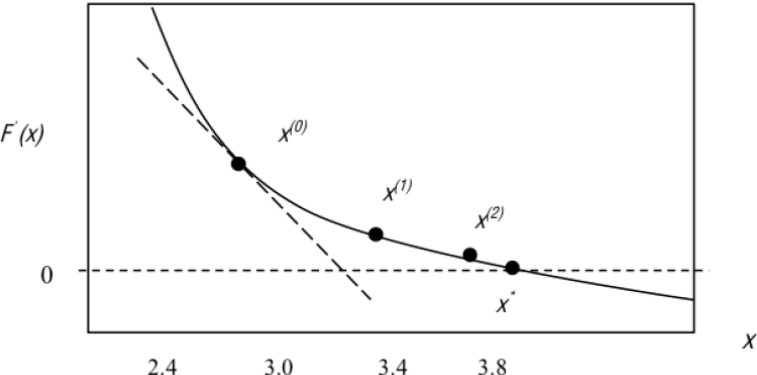

**Figure 1.** The Newton method of convergence.

## 4. Empirical Results

*4.1. Data Description*

The future prices data considered in this study consisted of five agricultural commodities including cocoa, coffee, corn, soybeans and wheat. Daily data were collected as 5000 observations between 2000 and 2020. Basic information consisted of the mean value, maximum and minimum value and standard deviation, as displayed in Table 2.

**Table 2.** Descriptive data of future prices for five agricultural commodities.

|          | Cocoa      | Coffee   | Corn      | Soybeans  | Wheat     |
|----------|------------|----------|-----------|-----------|-----------|
| Mean     | 2167.789   | 122.8145 | 382.1173  | 930.8092  | 502.5286  |
| Median   | 2205.000   | 118.6000 | 361.7500  | 945.1900  | 480.6300  |
| Max.     | 3774.000   | 304.9000 | 831.2500  | 1764.750  | 1280.000  |
| Min.     | 674.0000   | 41.50000 | 174.7500  | 418.0000  | 233.5000  |
| Std.Dev. | 701.7960   | 48.84470 | 160.7520  | 326.3169  | 183.2688  |
| Sum      | 10,838,947 | 614,072  | 1,910,587 | 4,654,046 | 2,512,643 |
| Obs.     | 5000       | 5000     | 5000      | 5000      | 5000      |

*4.2. Stationary Testing*

Empirically, the data from agricultural future prices are the time series data. Thus, the data should be tested for the stationary data. In this paper, the unit root test based on the Bayesian method is used to investigate the stationary data which is shown in Table 3. The null hypothesis (H0) is non-stationary and the alternative hypothesis (H1) is stationary. The results show that all of the time-series data from agricultural future prices are non-stationary or (I(1)).

**Table 3.** The unit root test relies on the Bayesian inference in daily data of five agricultural commodity future prices.

| Agricultural Commodities | Bayesian Factor Ratios (M1/M2) | Interpretation | Result |
|--------------------------|--------------------------------|----------------|--------|
| Cocoa    | 1005 | Strong evidence for Ni | I(1) |
| Coffee   | 1076 | Strong evidence for Ni | I(1) |
| Corn     | 1048 | Strong evidence for Ni | I(1) |
| Soybeans | 1031 | Strong evidence for Ni | I(1) |
| Wheat    | 1114 | Strong evidence for Ni | I(1) |

*4.3. Estimation of Extreme Value Using Bayesian Inference and the Newton Method*

From the result of Table 4, the extreme value estimation from the Bayesian inference estimation by using Non-stationary Extreme Value Analysis (NEVA) can be used to estimate both the non-stationary and stationary data. In this paper, all of the agricultural future

prices data are non-stationary, which the results deduced from this approach as shown in the interval value. Empirically, the interval of the extreme value from the Cocoa future prices is 2500 to 3500 USD/MT. For coffee future prices, the extreme value interval is between 180 and 220 USD/lb. Additionally, For the length of the interval extreme value for corn, soybeans and wheat future prices, the non-stationary estimated outcomes are shown as 440 to 480, 1300 to 1800, and 700 to 900, respectively. This is displayed in detail in Table 4.

**Table 4.** The optimally extreme value calculation of five agricultural commodity future prices.

| Agricultural Commodity | Bayesian Extreme Estimation as Interval Value | General Mean | The Newton Method |
|---|---|---|---|
| Cocoa | 2500–3500 | 2908.6 | 3040.053 |
| Coffee | 180–220 | 204.6 | 197.0472 |
| Corn | 440–480 | 446.5 | 466.0243 |
| Soybeans | 1300–1800 | 1606.5 | 1684.319 |
| Wheat | 700–900 | 816.9 | 853.9703 |

The Newton optimization method is the efficiently computational tool that extends the details of the results from the NEVA to provide the optimal point at which an extreme value should be considered to be alarming. Empirically, in this paper, the general mean and Newton optimization method are employed; the Newton method can investigate the results with more detail and reliability than the general mean method. From the details in Table 4, the estimation of the general mean shows that the optimal point for cocoa, coffee, corn, soybeans and wheat future prices are 2908.6, 204.6, 446.5, 1606.5, and 816.9, respectively. However, the results from the Newton optimization method differ from the general mean method, which can be described as follows: the optimal point of the cocoa future price is 3040.053, the coffee future price is 197.0472, the corn future price is 1684.319, the soybeans future price is 1684.319 and the wheat future price is 853.9703. Additionally, the results of the Bayesian extreme estimation and Newton optimization method are shown in Table 4, Figures 2 and 3 below.

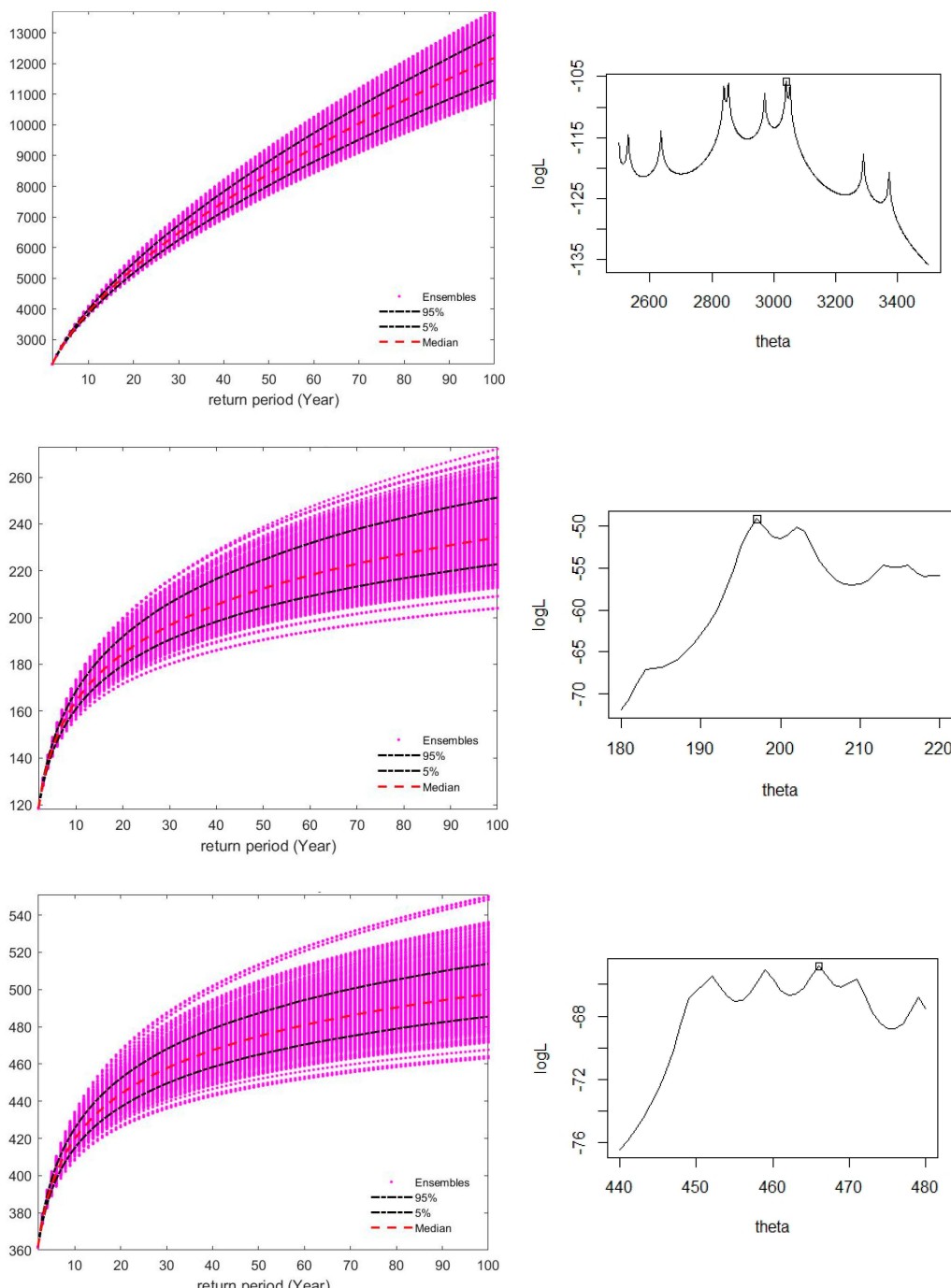

**Figure 2.** Presentation the Bayesian extreme results and the Newton-optimal point regarding cocoa, coffee, and corn.

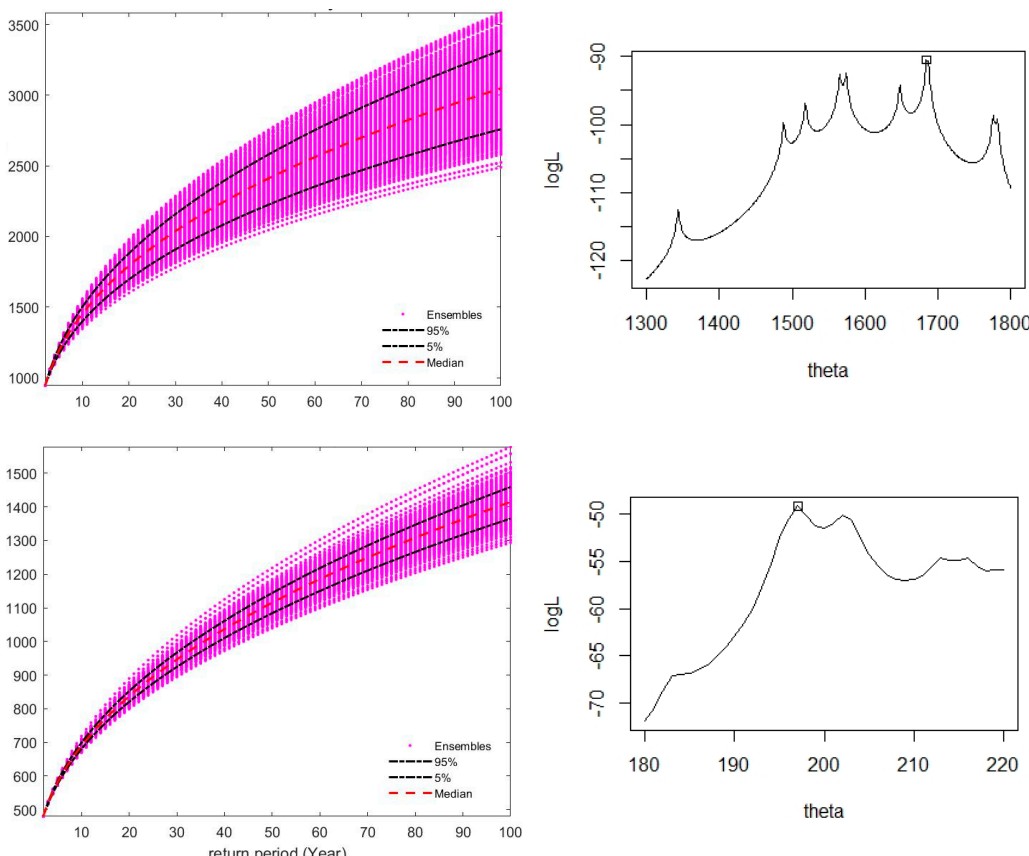

**Figure 3.** Presentation the Bayesian extreme results and the Newton-optimal point regarding soybeans and wheat.

## 5. Conclusions

Extreme events of five agricultural commodity future prices including cocoa, coffee, corn, soybeans and wheat as the most crucial major commodities in the United States are undergoing major changes and impacting other commodities in terms of agricultural markets and the economy. Here, time-series data were collected daily from 2000 to 2020 (5000 observations). Non-stationary Extreme Value Analysis (NEVA) using Bayesian inference, random variable processing and the Newton-optimal method were employed to determine the optimal point of extreme events. The NEVA method showed the interval value of extreme events from the measurement of agricultural commodity future prices. Results can be used to forecast the alarming interval value before the crisis or extreme event occurs. Furthermore, random variable processing and the Newton-optimal method were used to clarify more precise details of the optimal position exact point to define the highest value of agricultural commodity prices. Results from the Newton optimization can be used to forecast warning prices for farmers and investors as follows: the optimal point of cocoa future price was determined as 3040.053, coffee 197.0472, corn 1684.319, soybeans future price is 1684.319 and wheat 853.9703. Investors should prepare to sell their contract before the future prices of these agricultural commodities decrease. Agriculturists can use this information as advanced warning of alarming points of agricultural commodity prices to predict the most efficient quantity of their agricultural product to sell, with better ways to manage this risk.

The estimation of results using the Non-stationary Extreme Value Analysis (NEVA) method with Bayesian inference and the Newton-optimal method can support policy-makers to make decisions to prevent a crisis in the agricultural market and prepare solutions to solve this problem. Our results, showing computed interval values and optimal points,

can provide useful additional information about extreme events. Bayesian interference is an important computational statistical method for econometrics research.

**Author Contributions:** Conceptualization, C.C.; Data curation, J.S.; Methodology, C.C. and J.S.; Writing—original draft, J.S.; Writing—review & editing, S.S. All authors have read and agreed to the published version of the manuscript.

**Funding:** This research received no external funding.

**Data Availability Statement:** The data used in this paper are available from the investing (https://th.investing.com) at 25 May 2020.

**Acknowledgments:** This work was supported by Chiang Mai University, Thailand.

**Conflicts of Interest:** The authors declare no conflict of interest.

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
