# Peer review of "The Optimization of Bayesian Extreme Value: Empirical Evidence for the Agricultural Commodities in the US"

_economies, doi:10.3390/economies9010030_

Round 1

Reviewer 1 Report

Although it is an important topic which has received little attention in literature, the paper is ill written and it suffers from severe methodological limitations and a scarce theoretical framework.

The absence of a sound conceptual ground is revealed by the literature section which is missing. There are so little explicit matches between the few mentions of agricultural commodity futures prices in the United State including cocoa, coffee, corn, soybeans, and wheat and the literature on optimization of bayesian extreme value specific features.

 There is only a weak connection between the findings and the conclusion section.

Author Response

Dear Reviewer

I have already revised my research article to follow your suggestions

Thank you,

Reviewer 2 Report

1. The research data collects daily data during 2000-2020. However, during this period, it happened to encounter the 2008 financial tsunami and the recent US-China trade war. Although the estimation of the extreme value of the subject matter can be explained by the crisis or unusual time series trends in the financial data, as well as supporting investors and agricultural science The best point to make a decision and understand the warning level of agricultural futures prices. However, the structural variables of the financial crisis and the trade war are different. It is questionable whether the extreme responses of the two can be compared together?

2. Data attributes are verified through the ADF unit root test, can the author further verify the nonlinearity through KPSS test?

3. The interpretation of the Bayes factor in Table 1 should further explain why the ranges are classified in this way.

4. In Equation 14, X0 is the initial value. Is there a fixed range?

5. This study provides the best interval value through the results of Bayesian inference and Newton's best method of non-stationary extreme value analysis (NEVA). And further proves that the results of Newton-optimal method show that the extreme point of futures prices is the local highest point, and market participants can be notified to sell contracts and products before the commodity price index drops. However, Newton-optimal method is only through calculation and comparison. Can the performance difference be further tested through statistics?

6. It is suggested that the author can provide more relevant method application literature to support and echo this research results. 

Author Response

Dear Reviewer

I have already revised my research article to follow your suggestion.

Thank you,

  1. The research data collects daily data during 2000-2020. However, during this period, it happened to encounter the 2008 financial tsunami and the recent US-China trade war. Although the estimation of the extreme value of the subject matter can be explained by the crisis or unusual time series trends in the financial data, as well as supporting investors and agricultural science the best point to make a decision and understand the warning level of agricultural futures prices. However, the structural variables of the financial crisis and the trade war are different. It is questionable whether the extreme responses of the two can be compared together?

Response 1: Thank you for your comment. Yes, it cannot be used to compare together. So, I cut the term of trade war off to avoid some confusion.

  1. Data attributes are verified through the ADF unit root test, can the author further verify the nonlinearity through KPSS test?

Response 2: Due to the unit root test using Bayesian, this method only provides the ADF root test for test the stationary of data in term of Bayesian. Therefore, I cannot estimate unit root test of KPSS using Bayesian.

  1. The interpretation of the Bayes factor in Table 1 should further explain why the ranges are classified in this way.

Response 3: This interpretation of Bayes factor is referred from the Jeffrey Guideline model, but I did some mistake so some contents was missing. Therefore, I revised it regarding the explanation of Bayes factor of Jeffrey Guideline model.

  1. In Equation 14, X0is the initial value. Is there a fixed range?

Response 4: Yes, it is a fixed range that according to the linear approximation of :  and I added more interpretation of equation 14.

  1. This study provides the best interval value through the results of Bayesian inference and Newton's best method of non-stationary extreme value analysis (NEVA). And further proves that the results of Newton-optimal method show that the extreme point of futures prices is the local highest point, and market participants can be notified to sell contracts and products before the commodity price index drops. However, Newton-optimal method is only through calculation and comparison. Can the performance difference be further tested through statistics?

Response 5: For the result of newton-optimal method aim to find more accuracy of the empirical result after estimate the non-stationary extreme value analysis (NEVA) since the result from the NEVA provide only the range of local highest value which is the interval estimation.  Therefore, the newton method is used to find more detail of the local highest point but it can give more accurate result of mean than general mean.

  1. It is suggested that the author can provide more relevant method application literature to support and echo this research results. 

Response 6: There is the limitation of study on the analysis the nonstationary extreme value analysis (NEVA) applying Bayesian inference with the financial data and commodities price, especially in agricultural commodities. However, I provide most of the previous study which use the similar method to applied in the other topics.

Round 2

Reviewer 1 Report

The author(s) have addressed insufficient to my recommendations. Still, the author(s) needs to provide additional revisions regarding:

The absence of a sound conceptual ground is revealed by the literature section which is missing. The author(s) haven not inserted this section despite the fact I have made this recommendation. Also, there are so little explicit matches between the few mentions of agricultural commodity futures prices in the United State including cocoa, coffee, corn, soybeans, and wheat and the literature on optimization of bayesian extreme value specific features. 

The author(s) needs to provide satisfactory revisions in this line. 

Author Response

Dear Editor 

According to your requested, I have already done please check it from the attached file.

Thank you so much.

Round 3

Reviewer 1 Report

the author(s) have already revised my research article to follow my suggestions

Author Response

Dear reviewer
According to your suggestion, I have already revised follow your suggestion.
Please consider it from attached file
Sincerely.